# ADADQH OPTIMIZER: EVOLVING FROM STOCHASTIC TO ADAPTIVE BY AUTO SWITCH OF PRECONDITION MATRIX

## ABSTRACT

Adaptive optimizers (e.g., Adam) have achieved tremendous success in deep learning. The key component of the optimizer is the precondition matrix, which provides more gradient information and adjusts the step size of each gradient direction. Intuitively, the closer the precondition matrix approximates the Hessian, the faster convergence and better generalization the optimizer can achieve in terms of iterations. However, this performance improvement is usually accompanied by a huge increase in the amount of computation. In this paper, we propose a new optimizer called AdaDQH to achieve better generalization with acceptable computational overhead. The intuitions are the trade-off of the precondition matrix between computation time and approximation of Hessian, and the auto switch of the precondition matrix from Stochastic Gradient Descent (SGD) to the adaptive optimizer. We evaluate AdaDQH on public datasets of Computer Vision (CV), Natural Language Processing (NLP) and Recommendation Systems (RecSys). The experimental results reveal that, compared to the State-Of-The-Art (SOTA) optimizers, AdaDQH can achieve significantly better or highly competitive performance. Furthermore, we analyze how AdaDQH is able to auto switch from stochastic to adaptive and the actual effects in different scenes. The code is available in the supplemental material.

## 1 INTRODUCTION

Consider the following empirical risk minimization problems:

$$\min_{\boldsymbol{w} \in \mathbb{R}^n} f(\boldsymbol{w}) := \frac{1}{M} \sum_{k=1}^{M} \ell(\boldsymbol{w}; \boldsymbol{x}_k), \tag{1}$$

where $\boldsymbol{w} \in \mathbb{R}^n$ is a vector of parameters to be optimized, $\{\boldsymbol{x}_1, \dots, \boldsymbol{x}_M\}$ is a training set, and $\ell(\boldsymbol{w}; \boldsymbol{x})$ is a loss function measuring the performance of the parameter $\boldsymbol{w}$ on the example $\boldsymbol{x}$. Since it is ineffective to calculate the exact gradient in each optimization iteration when $M$ is large, we usually adopt a mini-batched stochastic gradient, which is

$$\boldsymbol{g}(\boldsymbol{w}) = \frac{1}{|\mathcal{B}|} \sum_{k \in \mathcal{B}} \nabla \ell(\boldsymbol{w}; \boldsymbol{x}_k),$$

where $\mathcal{B} \subset \{1, \dots, M\}$ is the sample set of size $|\mathcal{B}| \ll M$. Obviously, we have $\mathbf{E}_{p(\boldsymbol{x})}[\boldsymbol{g}(\boldsymbol{w})] = \nabla f(\boldsymbol{w})$ where $p(\boldsymbol{x})$ is the distribution of the training data. Equation 1 is usually solved iteratively. Assume $\boldsymbol{w}_t$ is already known and let $\Delta \boldsymbol{w} = \boldsymbol{w}_{t+1} - \boldsymbol{w}_t$, then

$$\begin{aligned}
\underset{\boldsymbol{w}_{t+1} \in \mathbb{R}^n}{\arg\min} f(\boldsymbol{w}_{t+1}) &= \underset{\Delta \boldsymbol{w} \in \mathbb{R}^n}{\arg\min} f(\Delta \boldsymbol{w} + \boldsymbol{w}_t) \\
&\approx \underset{\Delta \boldsymbol{w} \in \mathbb{R}^n}{\arg\min} f(\boldsymbol{w}_t) + (\Delta \boldsymbol{w})^T \nabla f(\boldsymbol{w}_t) + \frac{1}{2} (\Delta \boldsymbol{w})^T \nabla^2 f(\boldsymbol{w}_t) \Delta \boldsymbol{w} \\
&\approx \underset{\Delta \boldsymbol{w} \in \mathbb{R}^n}{\arg\min} \underbrace{f(\boldsymbol{w}_t) + (\Delta \boldsymbol{w})^T \nabla f(\boldsymbol{w}_t) + \frac{1}{2} (\Delta \boldsymbol{w})^T B_t \Delta \boldsymbol{w}}_{h(\Delta \boldsymbol{w})},
\end{aligned} \tag{2}$$

where the first approximation is from Taylor expansion. By solving Equation 2 and using $\boldsymbol{m_t}$ to replace $\nabla f(\boldsymbol{w_t})$, the general update formula is

$$\boldsymbol{w}_{t+1} = \boldsymbol{w}_t - \alpha_t B_t^{-1} \boldsymbol{m}_t, \quad t \in \{1, 2, \ldots, T\}, \tag{3}$$

where $\alpha_t$ is the step size for avoiding divergence, $\boldsymbol{m}_t \approx \mathbf{E}_{p(\boldsymbol{x})}[\boldsymbol{g}_t]$ is the first moment term which is the weighted average of gradient $\boldsymbol{g}_t$ and $B_t$ is the so-called precondition matrix that incorporates additional information and adjusts updated velocity of variable $\boldsymbol{w}_t$ in each direction. Most of gradient descent algorithms can be summarized with Equation 3 such as SGD (Robbins & Monro, 1951), MOMENTUM (Polyak, 1964), ADAGRAD (Duchi et al., 2011), ADADELTA (Zeiler, 2012), ADAM (Kingma & Ba, 2015), AMSGRAD (Reddi et al., 2018), ADABELIEF (Zhuang et al., 2020) and ADAHESSIAN (Yao et al., 2020). Intuitively, the closer $B_t$ approximates the Hessian, the closer $h(\Delta \boldsymbol{w})$ approximates $f(\boldsymbol{w}_{t+1})$. Consequently, we can achieve a more accurate solution in terms of iterations. However, it is usually untrue in terms of runtime. For instance, ADAHESSIAN that approximates the diagonal Hessian consumes $2.91\times$ more computation time than ADAM for ResNet32 on Cifar10 (Yao et al., 2020). Therefore, the key factor of designing the precondition matrix is how to trade off the approximation degree of the Hessian and the computation complexity.

In this paper, we propose AdaDQH (**Ada**ptive optimizer with **D**iagonal **Q**uasi-**H**essian), whose precondition matrix is closely related to the Hessian but computationally efficient. Furthermore, AdaDQH can auto switch the precondition matrix from SGD to the adaptive optimizer through the hyperparameter threshold $\delta$. Our contributions can be summarized as follows.

- We propose AdaDQH, which originates the new design of the precondition matrix. We establish theoretically proven convergence guarantees in both convex and non-convex stochastic settings.

- We validate AdaDQH on a total of six public datasets: two from CV (Cifar10 (Krizhevsky et al., 2009) and ImageNet (Russakovsky et al., 2015)), two from NLP (IWSLT14 (Cettolo et al., 2014) and PTB (Marcus et al., 1993)) and the rest from RecSys (Criteo (Criteo, 2014) and Avazu (Avazu, 2015)). The experimental results reveal that AdaDQH can outperform or be on a par with the SOTA optimizers.

- We analyze how AdaDQH is able to auto switch from stochastic to adaptive, and assess the rigorous effect of the hyperparameter $\delta$ which controls the auto-switch process in different scenes.

RELATED WORK

By choosing different $B_t$ and $\boldsymbol{m}_t$ of Equation 3, different optimizers are invented from the standard second order optimizer, i.e., Gauss-Newton method to the standard first order optimizer, i.e., SGD where $\boldsymbol{m}_t$ is usually designed for noise reduction and $B_t$ for solving the ill-conditioned problems. See Table 1. Kunstner et al. (2019) shows that the Fisher information matrix can be the reasonable approximation of the Hessian whereas the empirical Fisher can't. Furthermore, they propose the concept of variance adaption to explain the practical success of the empirical Fisher preconditioning.

The hybrid optimization methods of switching an adaptive optimizer to SGD have been proposed for improving the generalization performance, such as ADABOUND (Luo et al., 2019) and SWATS (Keskar & Socher, 2017). Luo et al. (2019) adopts clipping on the learning rate of ADAM, whose upper and lower bounds are a non-increasing and non-decreasing functions, respectively, which would converge to the learning rate of SGD. The clipping method is also mentioned in Keskar & Socher (2017), whose upper and lower bounds are constants.

NOTATION

We use lowercase letters to denote scalars, boldface lowercase to denote vectors, and uppercase letters to denote matrices. We denote a sequence of vectors by subscripts, that is, $\boldsymbol{x}_1, \ldots, \boldsymbol{x}_t$ where $t \in [T] := \{1, 2, \ldots, T\}$, and entries of each vector by an additional subscript, e.g., $x_{t,i}$. For any vectors $\boldsymbol{x}, \boldsymbol{y} \in \mathbb{R}^n$, we write $\boldsymbol{x}^T \boldsymbol{y}$ or $\boldsymbol{x} \cdot \boldsymbol{y}$ for the standard inner product, $\boldsymbol{xy}$ for element-wise multiplication, $\boldsymbol{x}/\boldsymbol{y}$ for element-wise division, $\sqrt{\boldsymbol{x}}$ for element-wise square root, $\boldsymbol{x}^2$ for element-wise square. For the standard Euclidean norm, $\|\boldsymbol{x}\| = \|\boldsymbol{x}\|_2 = \sqrt{\langle \boldsymbol{x}, \boldsymbol{x} \rangle}$ and $\max(\boldsymbol{x}, \boldsymbol{y})$ for element-wise maximum. We also use $\|\boldsymbol{x}\|_\infty = \max_i |x^{(i)}|$ to denote $\ell_\infty$-norm, where $x^{(i)}$ is the $i$-th element of $\boldsymbol{x}$. Let $\boldsymbol{e}_i$ denote the unit vector where the $i$-th element is one and $\nabla_i f$ denote the $i$-th element of $\nabla f$.

Table 1: Different optimizers with choosing different $B_t$.

| $B_t$ | Optimizer |
|---|---|
| $B_t = \mathbf{H}$ | GAUSS-HESSIAN |
| $B_t \approx \mathbf{H}$ | LBFGS (Byrd et al., 1995) |
| $B_t \approx \text{diag}(\mathbf{H})$ | ADAHESSIAN (Yao et al., 2020) |
| $B_t = \mathbf{F}$ | NATURAL GRADIENT (Amari, 1998) |
| $B_t^2 \approx \mathbf{F}_{emp}$ | SHAMPOO (Gupta et al., 2018) |
| $B_t^2 \approx \text{diag}(\mathbf{F}_{emp})$ | ADAGRAD (Duchi et al., 2011), ADADELTA (Zeiler, 2012), ADAM (Kingma & Ba, 2015), ADAMW (Loshchilov & Hutter, 2019), AMSGRAD (Reddi et al., 2018) |
| $B_t^2 \approx \text{diag}(\mathbf{Var}(\boldsymbol{g}_t))$ | ADABELIEF (Zhuang et al., 2020) |
| $B_t = \mathbb{I}$ | SGD (Robbins & Monro, 1951), MOMENTUM (Polyak, 1964) |

$\mathbf{H}$ is the Hessian. $\mathbf{F}$ is the Fisher information matrix. $\mathbf{F}_{emp}$ is the empirical Fisher information matrix.

Let $f_t(\boldsymbol{w})$ be the loss function of the model at $t$-step where $\boldsymbol{w} \in \mathbb{R}^n$. We consider $\boldsymbol{m}_t$ as Exponential Moving Averages (EMA) of $\boldsymbol{g}_t$ throughout this paper, i.e.,

$$\boldsymbol{m}_t = \beta_1 \boldsymbol{m}_{t-1} + (1 - \beta_1)\boldsymbol{g}_t = (1 - \beta_1)\sum_{i=1}^{t} \boldsymbol{g}_{t-i+1}\beta_1^{i-1}, \ t \geq 1, \tag{4}$$

where $\beta_1 \in [0, 1)$ is the exponential decay rate.

## 2 ALGORITHM

### 2.1 DETAILS AND INTUITIONS OF ADADQH OPTIMIZER

The algorithm is listed in Algorithm 1. The design of AdaDQH comes from two intuitions: Hessian

---

**Algorithm 1** AdaDQH

1: **Input:** parameters $\beta_1$, $\beta_2$, $\delta$, $\boldsymbol{w}_1 \in \mathbb{R}^n$, step size $\alpha_t$, initialize $\boldsymbol{m}_0 = \mathbf{0}, \boldsymbol{b}_0 = \mathbf{0}$
2: **for** $t = 1$ **to** $T$ **do**
3:     $\boldsymbol{g}_t = \nabla f_t(\boldsymbol{w}_t)$
4:     $\boldsymbol{m}_t \leftarrow \beta_1 \boldsymbol{m}_{t-1} + (1 - \beta_1)\boldsymbol{g}_t$
5:     $\boldsymbol{s}_t = \begin{cases} \boldsymbol{m}_1/(1-\beta_1) & t = 1 \\ \boldsymbol{m}_t/(1-\beta_1^t) - \boldsymbol{m}_{t-1}/(1-\beta_1^{t-1}) & t > 1 \end{cases}$
6:     $\boldsymbol{b}_t \leftarrow \beta_2 \boldsymbol{b}_{t-1} + (1 - \beta_2)\boldsymbol{s}_t^2$
7:     $\boldsymbol{w}_{t+1} = \boldsymbol{w}_t - \alpha_t \frac{\sqrt{1-\beta_2^t}}{1-\beta_1^t} \frac{\boldsymbol{m}_t}{\max(\sqrt{\boldsymbol{b}_t}, \delta\sqrt{1-\beta_2^t})}$
8: **end for**

---

approximation and auto switch for fast convergence and good generalization across tasks.

HESSIAN APPROXIMATION     Let $\Delta \boldsymbol{w} = -\alpha_t B_t^{-1}\boldsymbol{m}_t$ of Equation 3, then we have

$$\begin{aligned}
\mathbf{E}[g_{t,i} - g_{t-1,i}] &= \nabla_i f(\boldsymbol{w}_t) - \nabla_i f(\boldsymbol{w}_{t-1}) \\
&= \nabla\nabla_i f(\boldsymbol{w}_{t-1} + \theta\Delta\boldsymbol{w}) \cdot \Delta\boldsymbol{w}, \ \theta \in (0, 1) \\
&\stackrel{\theta=1}{\approx} \nabla\nabla_i f(\boldsymbol{w}_t) \cdot \Delta\boldsymbol{w} \\
&\stackrel{\Delta\boldsymbol{w}=e_i}{\approx} \nabla_i\nabla_i f(\boldsymbol{w}_t),
\end{aligned} \tag{5}$$

where the second equality above follows from the mean value theorem and in the second approximation we assume that $\boldsymbol{w}_t$ is not updated except for the $i$-th direction. Therefore, we can see that $\mathbf{E}[\boldsymbol{g}(\boldsymbol{w}_t) - \boldsymbol{g}(\boldsymbol{w}_{t-1})]$ is closely related to $\text{diag}(\mathbf{H}(\boldsymbol{w}_t))$. Similar to Kingma & Ba (2015), we use

$m_t/(1 - \beta_1^t)$ to approximate $\mathbf{E}[g(w_t)]$ where the denominator is for bias correction. Denote

$$s_t = \begin{cases} m_1/(1 - \beta_1) & t = 1, \\ m_t/(1 - \beta_1^t) - m_{t-1}/(1 - \beta_1^{t-1}) & t > 1. \end{cases}$$

Therefore, we choose the precondition matrix $B_t$ satisfying

$$B_t^2 = \text{diag}(\text{EMA}(s_1 s_1^T, s_2 s_2^T, \cdots, s_t s_t^T))/(1 - \beta_2^t),$$

where $\beta_2$ is the parameter of EMA and the denominator is also for bias correction.

AUTO SWITCH  Normally, a small value is added to $B_t$ for numerical stability, becoming $B_t + \epsilon \mathbb{I}$. However, we replace it with $\max(B_t, \delta \mathbb{I})$, where we use a different notation $\delta$ to indicate its essential role in auto switch. When $\hat{b}_t := \sqrt{b_t/(1 - \beta_2^t)}$ is relatively larger than $\delta$, AdaDQH takes a confident step in the adaptive way. Otherwise, the update is EMA, i.e. $m_t$, with a constant scale $\alpha_t/(1 - \beta_1^t)$, similar to SGD with momentum. Moreover, AdaDQH can auto switch modes in a per parameter manner as training progresses.

Compared to the additive method, AdaDQH can eliminate the noise caused by $\epsilon$ in adaptive updates. Another major benefit is that AdaDQH has the ability to generalize in different tasks by tuning $\delta$, without choosing from candidates of obscure optimizers empirically. The effect of $\delta$ is discussed in Sec.3.5, experimentally.

## 2.2 CONVERGENCE ANALYSIS

Using the framework developed in Reddi et al. (2018); Yang et al. (2016); Chen et al. (2019); Zhou et al. (2018), we have the following theorems that provide the convergence in convex and non-convex settings. Particularly, we use $\beta_{1,t}$ to replace $\beta_1$ where $\beta_{1,t}$ is non-increasing.

**Theorem 1.** *(Convergence in convex settings) Let $\{w_t\}$ be the sequence obtained by AdaDQH (Algorithm 1), $\alpha_t = \alpha/\sqrt{t}$, $\beta_{1,t} \leq \beta_1 \in [0,1)$, $\beta_2 \in [0,1)$, $b_{t,i} \leq b_{t+1,i} \ \forall i \in [n]$ and $\|g_t\|_\infty \leq G_\infty, \forall t \in [T]$. Suppose $f_t(w)$ is convex for all $t \in [T]$, $w^*$ is an optimal solution of $\sum_{t=1}^T f_t(w)$, i.e., $w^* = \arg\min_{w \in \mathbb{R}^n} \sum_{t=1}^T f_t(w)$ and there exists the constant $D_\infty$ such that $\max_{t \in [T]} \|w_t - w^*\|_\infty \leq D_\infty$. Then we have the following bound on the regret*

$$\sum_{t=1}^T (f_t(w_t) - f_t(w^*)) < \frac{1}{1 - \beta_1} \left[ \frac{n(2G_\infty + \delta)D_\infty^2}{2\alpha\sqrt{1 - \beta_2}(1 - \beta_1)^2} \sqrt{T} + \sum_{t=1}^T \frac{\beta_{1,t}}{2\hat{\alpha}_t} nD_\infty^2 \right.$$
$$\left. + \frac{n\alpha G_\infty^2}{(1 - \beta_1)^3} \left( 1 + \frac{1}{\delta\sqrt{1 - \beta_2}} \right) \sqrt{T} \right].$$

The proof of Theorem 1 is given in Appendix A.

**Corollary 1.** *Suppose $\beta_{1,t} = \beta_1/t$, then we have*

$$\sum_{t=1}^T (f_t(w_t) - f_t(w^*)) < \frac{1}{1 - \beta_1} \left[ \frac{n(2G_\infty + \delta)D_\infty^2}{2\alpha\sqrt{1 - \beta_2}(1 - \beta_1)^2} \sqrt{T} + \frac{nD_\infty^2 \beta_1}{\alpha\sqrt{1 - \beta_2}} \sqrt{T} \right.$$
$$\left. + \frac{n\alpha G_\infty^2}{(1 - \beta_1)^3} \left( 1 + \frac{1}{\delta\sqrt{1 - \beta_2}} \right) \sqrt{T} \right].$$

The proof of Corollary 1 is given in Appendix B. Corollary 1 implies the regret is $O(\sqrt{T})$ and can achieve the convergence rate $O(1/\sqrt{T})$ in convex settings.

**Theorem 2.** *(Convergence in non-convex settings) Suppose that the following assumptions are satisfied:*

1. *$f$ is differential and lower bounded, i.e., $f(w^*) > -\infty$ where $w^*$ is an optimal solution. $f$ is also $L$-smooth, i.e., $\forall u, v \in \mathbb{R}^n$, we have*

$$f(u) \leq f(v) + \langle \nabla f(v), u - v \rangle + \frac{L}{2} \|u - v\|^2.$$

2. *At step t, the algorithm can access a bounded noisy gradient and the true gradient is bounded, i.e., $\|\boldsymbol{g}_t\|_\infty \leq G_\infty, \|\nabla f(\boldsymbol{w}_t)\|_\infty \leq G_\infty, \forall t \in [T]$. Without loss of generality, we assume $G_\infty \geq \delta$.*

3. *The noisy gradient is unbiased and the noise is independent, i.e., $\boldsymbol{g}_t = \nabla f(\boldsymbol{w}_t) + \boldsymbol{\zeta}_t, \mathbf{E}[\boldsymbol{\zeta}_t] = \mathbf{0}$ and $\boldsymbol{\zeta}_i$ is independent of $\boldsymbol{\zeta}_j$ if $i \neq j$.*

4. $\alpha_t = \alpha/\sqrt{t}$, $\beta_{1,t} \leq \beta_1 \in [0,1)$, $\beta_2 \in [0,1)$ and $b_{t,i} \leq b_{t+1,i} \forall i \in [n]$.

*Then Algorithm 1 yields*

$$\min_{t \in [T]} \mathbf{E}[\|\nabla f(\boldsymbol{w}_t)\|^2] < C_1 \frac{1}{\sqrt{T}-\sqrt{2}} + C_2 \frac{\log T}{\sqrt{T}-\sqrt{2}} + C_3 \frac{\sum_{t=1}^T \hat{\alpha}_t(\beta_{1,t} - \beta_{1,t+1})}{\sqrt{T}-\sqrt{2}},$$

*where $C_1$, $C_2$ and $C_3$ are defined as follows:*

$$C_1 = \frac{G_\infty}{\alpha(1-\beta_1)^2(1-\beta_2)^2}\left(f(\boldsymbol{w}_1) - f(\boldsymbol{w}^*) + \frac{nG_\infty^2 \alpha}{(1-\beta_1)^8 \delta^2}(\delta + 8L\alpha) + \frac{\alpha\beta_1 nG_\infty^2}{(1-\beta_1)^3 \delta}\right),$$

$$C_2 = \frac{15LnG_\infty^3 \alpha}{2(1-\beta_2)^2(1-\beta_1)^{10}\delta^2},$$

$$C_3 = \frac{nG_\infty^3}{\alpha(1-\beta_1)^5(1-\beta_2)^2 \delta}.$$

The proof of Theorem 2 is given in Appendix C. Note that we can let $\boldsymbol{b}_{t+1} = \max(\boldsymbol{b}_{t+1}, \boldsymbol{b}_t)$, which is usually called AMSGrad condition (Reddi et al., 2018), to make sure the assumption $b_{t,i} \leq b_{t+1,i} \forall i \in [n]$ always true, though it could degenerate the effect in practice. The more detailed analysis is given in Appendix E.3. From Theorem 2, we have the following corollaries.

**Corollary 2.** *Suppose $\beta_{1,t} = \beta_1/\sqrt{t}$, then we have*

$$\min_{t \in [T]} \mathbf{E}[\|\nabla f(\boldsymbol{w}_t)\|^2] < C_4 \frac{1}{\sqrt{T}-\sqrt{2}} + C_5 \frac{\log T}{\sqrt{T}-\sqrt{2}},$$

*where $C_4$ and $C_5$ are defined as follows:*

$$C_4 = \frac{G_\infty}{\alpha(1-\beta_1)^2(1-\beta_2)^2}\left(f(\boldsymbol{w}_1) - f(\boldsymbol{w}^*) + \frac{nG_\infty^2 \alpha}{(1-\beta_1)^8 \delta^2}(2\delta + 8L\alpha) + \frac{\alpha\beta_1 nG_\infty^2}{(1-\beta_1)^3 \delta}\right),$$

$$C_5 = \frac{nG_\infty^3}{(1-\beta_2)^2(1-\beta_1)^{10}\delta^2}\left(\frac{15}{2}L\alpha + \delta\right).$$

The proof of Corollary 2 is given in Appendix D.

**Corollary 3.** *Suppose $\beta_{1,t} = \beta_1$, $\forall t \in [T]$, then we have*

$$\min_{t \in [T]} \mathbf{E}[\|\nabla f(\boldsymbol{w}_t)\|^2] < C_1 \frac{1}{\sqrt{T}-\sqrt{2}} + C_2 \frac{\log T}{\sqrt{T}-\sqrt{2}},$$

*where $C_1$ and $C_2$ are the same with Theorem 2.*

Corollaries 2, 3 imply the convergence (to the stationary point) rate for AdaDQH is $O(\log T/\sqrt{T})$ in non-convex settings.

## 2.3 NUMERICAL ANALYSIS

In this section, we compare AdaDQH against several SOTA optimizers in three test funtions. We adopt the parameter settings from Zhuang et al. (2020). The learning rate is set to 1e-3 for all adaptive optimizers, along with the same epsilon/delta (1e-8), betas ($\beta_1 = 0.9, \beta_2 = 0.999$). For SGD, momentum is set to 0.9 and learning rate is 1e-6 for numerical stability. AdaDQH shows promising results as it reaches the optimal points across all of the experiments, shown in Figure 1. Furthermore, we search the best leaning rate for each optimizer with regard to Beale function. AdaDQH is still the strongest competitor. The details are provided in E.1.

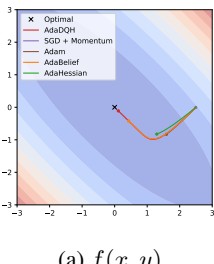 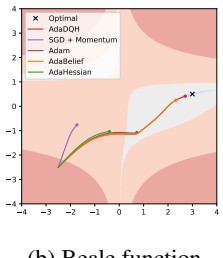 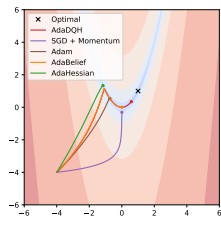

(a) $f(x, y)$         (b) Beale function         (c) Rosenbrock function

Figure 1: Trajectories of different optimizers in three test functions, where $f(x, y) = (x + y)^2 + (x - y)^2/10$. We also provide animated versions in the supplemental material.

Table 2: Experiments overview

| Task | Dataset | Model | *Train* | *Val/Test* | *Params* |
|------|---------|-------|-------|----------|--------|
| CV | Cifar10 | ResNet20/ResNet32 | 50K | 10K | 0.27M/0.47M |
| | ImageNet | ResNet18 | 1.28M | 50K | 11.69M |
| NLP-NMT | IWSLT14 De-En | Transformer small | 153K | 7K/7K | 36.7M |
| | | 1-layer LSTM | | | 5.3M |
| NLP-LM | PTB | 2-layer LSTM | 0.93M | 730K/82K | 13.6M |
| | | 3-layer LSTM | | | 24.2M |
| RecSys | Avazu | MLP | 36.2M | 4.2M | 151M |
| | Criteo | DCN | 39.4M | 6.6M | 270M |

## 3 EXPERIMENTS

### 3.1 EXPERIMENT SETUP

We experimentally compare the performance of different optimizers on a wide range of learning tasks, including CV, NLP and RecSys. The details of the tasks are as follows.

**CV:** We experiment with ResNet20 and ResNet32 on Cifar10 (Krizhevsky et al., 2009) dataset, and ResNet18 on ImageNet (Russakovsky et al., 2015) dataset. The details of the datasets are listed in Table 2. We train 160 epochs and decay the learning rate by a factor of 10 at epoch 80 and 120 for Cifar10, and train 90 epochs and decay the learning rate by a factor of 10 every 30 epochs for ImageNet. The batch size is 256 for both datasets.

**NLP:** We experiment with Neural Machine Translation (NMT) on IWSLT14 German-to-English (De-En) (Cettolo et al., 2014), and Language Modeling (LM) on Penn TreeBank (Marcus et al., 1993). For NMT task, transformer small architecture is adopted. We use the same setting and pre-processing method in Yao et al. (2020), as well as the same length penalty (1.0), beam size (5) and max tokens (4096). We train 55 epochs and average the last 5 checkpoints for inference. For LM task, we train 1,2,3-layer LSTM with batch size of 20 for 200 epochs. The details are listed in Table 2. Additionally, we keep settings like learning rate scheduler and warm-up steps identical for the same task.

**RecSys:** We experiment on two common datasets including Avazu (Avazu, 2015) and Criteo (Criteo, 2014) which are various display ads logs for the purpose of predicting the Click Through Rate (CTR). For Avazu, the samples from the first nine days are used for training, while the rest are for testing. We use the basic Multilayer Perceptron (MLP) structure of most deep CTR models. Specifically, the model maps each categorical feature as a 16-dimensional embedding vector, following up with 4 fully connected layer of dimension in 64,32,16,1, respectively. For Criteo, we take the early 6/7 part of all samples as the train set. We adopt Deep & Cross Network (DCN) (Wang et al., 2017) with embedding size set to 8, along with 2 deep layers of size 64 and 2 cross layers. The details are listed in Table 2. We train 1 epoch with a batch size of 512 for both datasets.

Optimizers to be compared include SGD (Robbins & Monro, 1951), Adam (Kingma & Ba, 2015), AdamW (Loshchilov & Hutter, 2019), AdaBelief (Zhuang et al., 2020) and AdaHessian (Yao et al., 2020). The choices of the hyperparameters are given in Appendix E.2. The experiments of CV and

Table 3: Top-1 accuracy for different optimizers when trained on Cifar10 and ImageNet.

| Dataset | Cifar10 | | ImageNet |
|---|---|---|---|
| Model | ResNet20 | ResNet32 | ResNet18 |
| SGD | $92.14 \pm .14$ | $93.10 \pm .07$ | $69.85 \pm .04$ |
| Adam | $90.46 \pm .20$ | $91.54 \pm .12$ | $63.81 \pm .26$ |
| AdamW | $92.12 \pm .14$ | $92.72 \pm .01$ | $68.91 \pm .09$ |
| AdaBelief | $92.19 \pm .15$ | $92.90 \pm .13$ | $69.93 \pm .09$ |
| AdaHessian | $92.27 \pm .27$ | $92.91 \pm .14$ | $69.94 \pm .09$ |
| AdaDQH | $\mathbf{92.35 \pm .24}$ | $\mathbf{93.12 \pm .18}$ | $\mathbf{70.19 \pm .05}$ |

Table 4: Relative training time for AdaDQH (baseline), SGD and AdaHessian. Additionally, minutes of training one epoch with AdaDQH are provided. † is measured on one *Nvidia P100 GPU*, §/‡ on one/four *Nvidia V100 GPU*. Note that * results from the limitations of PyTorch running RNN model with second order optimizers.

| Dataset | Cifar10 | | ImageNet | IWSLT14 | PTB |
|---|---|---|---|---|---|
| Model | ResNet20 † | ResNet32 † | ResNet18 ‡ | Transformer † | 2-layer LSTM § |
| SGD | $0.85\times$ | $0.84\times$ | $0.62\times$ | $0.67\times$ | $0.93\times$ |
| AdaHessian | $1.86\times$ | $2.16\times$ | $2.98\times$ | $2.17\times$ | $9.80\times^*$ |
| AdaDQH | $1.00\times$ | $1.00\times$ | $1.00\times$ | $1.00\times$ | $1.00\times$ |
| AdaDQH (min/epoch) | 0.52 | 0.62 | 11.25 | 3.88 | 0.30 |

NLP are conducted with GPUs in the PyTorch framework (Paszke et al., 2019), and the experiments of RecSys are conducted with 3 parameter servers and 5 workers in the TensorFlow framework (Abadi et al., 2016). We run all the experiments 5 times with random seeds and calculate the statistical results.

## 3.2 CV

Table 3 reports the top-1 accuracy for different optimizers when trained on Cifar10 and ImageNet. It is remarkable that AdaDQH outperforms other optimizers on both Cifar10 and ImageNet. The testing accuracy ($[\mu \pm \sigma]$) curves of different optimizers for ResNet20/32 on Cifar10 and ResNet18 on ImageNet are plotted in Figure 2. Note that the results of SGD, AdaBelief and AdaHessian on ImageNet are lower than the number reported in original papers (Chen et al., 2020; Zhuang et al., 2020; Yao et al., 2020) which are run single time. More discussions are given in Appendix E.4. We also report the accuracy of AdaDQH for ResNet18 on Cifar10 for comparing with the SOTA results [1], which is listed in Appendix E.5. In addition, it is worth mentioning that although AdaDQH is considered as quasi-Hessian, its runtime cost is comparable with SGD and much lower than AdaHessian, which is shown in Table 4.

## 3.3 NLP

We report case-insensitive BiLingual Evaluation Understudy (BLEU, higher is better) score and the perplexity (PPL, lower is better) on test set for NMT and LM tasks respectively. The results are shown in Table 5. For the NMT task on IWSLT14, AdaDQH achieves a similar result to AdaBelief, outperforming the other optimizers. For LM task on PTB, AdaDQH obtains lowest PPL in all 1,2,3-layer LSTM experiments, as demonstrated in Figure 3. Furthermore, we report the relative training time in Table 4 which is similar to CV.

## 3.4 RecSys

We adopt Area Under the receiver-operator Curve (AUC) as the evaluation criterion which is a good measurement in CTR estimation (Graepel et al., 2010). Table 6 shows that compared to other

---

[1]https://paperswithcode.com/sota/stochastic-optimization-on-cifar-10-resnet-18

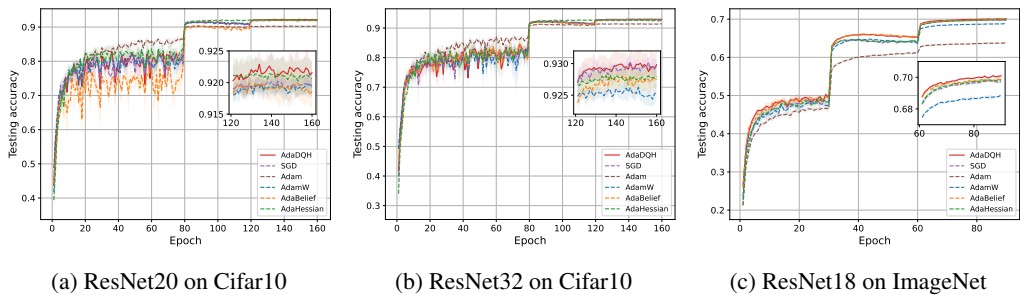

Figure 2: Testing accuracy curves of different optimizers for ResNet20/32 on Cifar10 and ResNet18 on ImageNet. The solid line represents the mean of the results and the shaded area represents the standard deviation.

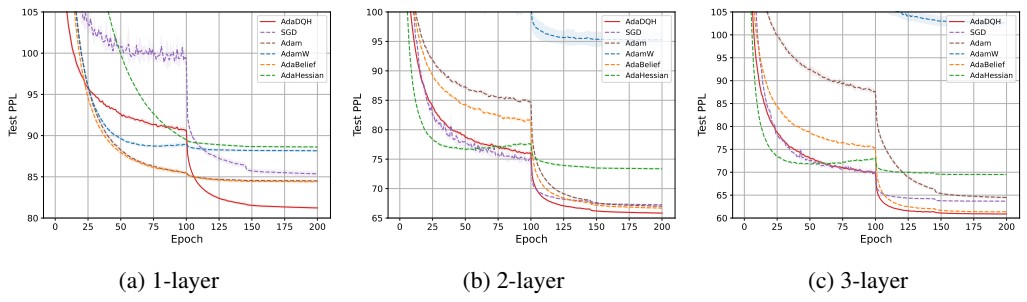

Figure 3: Test PPL ($[\mu \pm \sigma]$) on Penn Treebank for 1,2,3-layer LSTM.

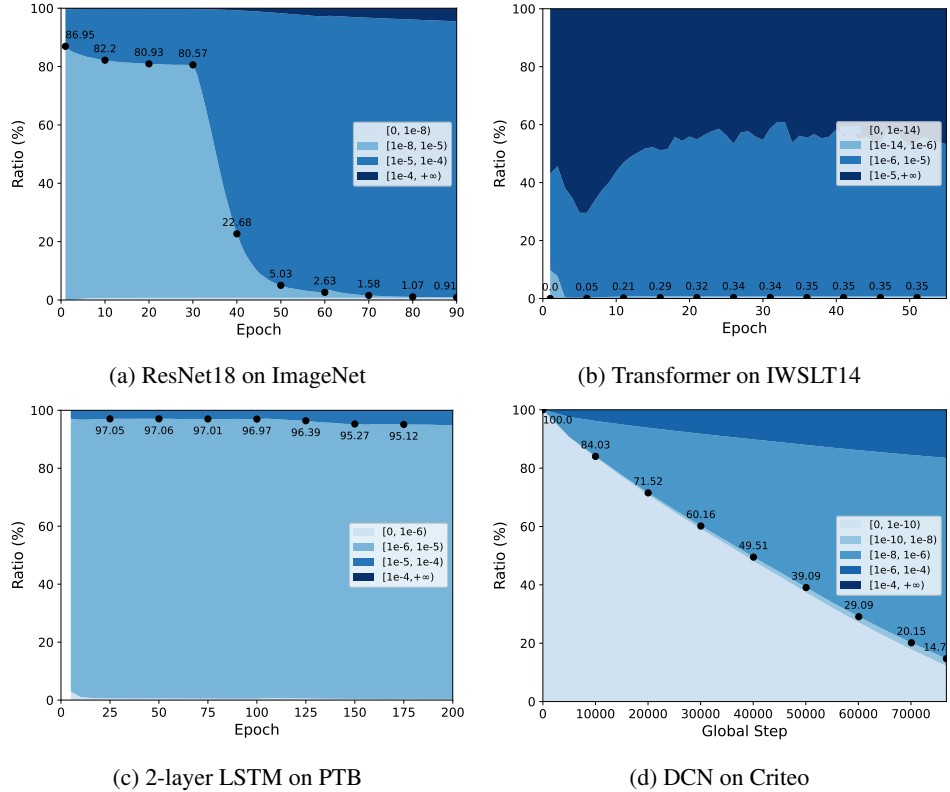

Figure 4: The distribution of $\hat{\boldsymbol{b}}_t$ on different epochs/steps. The colored area denotes the ratio of $\hat{\boldsymbol{b}}_t$ in the corresponding interval. The values of $\delta$ of ResNet18 on ImageNet, Transformer on IWSLT14, 2-layer LSTM on PTB, and DCN on Criteo are 1e-5, 1e-14, 1e-5 and 1e-8 respectively.

Table 5: Test BLEU score and PPL for NMT and LM tasks. † is reported in Yao et al. (2020).

| Dataset | IWSLT14 | PTB | | |
|---|---|---|---|---|
| Metric | *BLEU*, higher is better | PPL, lower is better | | |
| Model | Transformer | 1-layer LSTM | 2-layer LSTM | 3-layer LSTM |
| SGD | $28.57 \pm .15$ † | $85.36 \pm .34$ | $67.26 \pm .17$ | $63.68 \pm .17$ |
| Adam | $29.61 \pm .17$ | $84.50 \pm .16$ | $67.01 \pm .11$ | $64.45 \pm .26$ |
| AdamW | $35.75 \pm .03$ | $88.16 \pm .19$ | $95.25 \pm 1.33$ | $102.61 \pm 1.13$ |
| AdaBelief | $\mathbf{35.93 \pm .08}$ | $84.40 \pm .21$ | $66.69 \pm .23$ | $61.34 \pm .11$ |
| AdaHessian | $35.79 \pm .06$ † | $88.62 \pm .15$ | $73.37 \pm .22$ | $69.51 \pm .19$ |
| **AdaDQH** | $\mathbf{35.94 \pm .11}$ | $\mathbf{81.23 \pm .17}$ | $\mathbf{65.84 \pm .18}$ | $\mathbf{60.89 \pm .09}$ |

Table 6: Test AUC for different optimizers when trained on Avazu and Criteo.

| Dataset | Avazu | Criteo |
|---|---|---|
| Model | MLP | DCN |
| SGD | $0.7463 \pm .0005$ | $0.7296 \pm .0067$ |
| Adam | $0.7458 \pm .0010$ | $\mathbf{0.8023 \pm .0002}$ |
| AdaBelief | $0.7467 \pm .0009$ | $0.8022 \pm .0002$ |
| AdaHessian | $0.7434 \pm .0006$ | $0.7995 \pm .0018$ |
| AdaDQH | $\mathbf{0.7480 \pm .0008}$ | $\mathbf{0.8023 \pm .0004}$ |

optimizers, AdaDQH can achieve significantly better or highly competitive performance on the AUC metric.

## 3.5 THE EFFECT OF $\delta$

In this section, we analyze the rigorous effect of $\delta$, i.e., what exact percentage of $\hat{\boldsymbol{b}}_t$ is truncated by $\delta$ in Algorithm 1. Figure 4 depicts the distribution of $\hat{\boldsymbol{b}}_t$ during the training process on different tasks in the best configuration we found. The black dot gives the exact percentage of $\hat{\boldsymbol{b}}_t$ that is truncated by $\delta$ in the task. Lower percentage means more SGD-like updates than adaptive steps, which is controlled by the choice of $\delta$.

Figure 4 reveals how auto switch in AdaDQH works in different tasks. As shown in Figure 4a, AdaDQH behaves more like SGD in early stage of training (before the first learning rate decay at the 30th epoch) and switches to the adaptive for fine-tuning, since SGD generally outperforms the adaptive optimizers in CNN tasks. Figure 4b indicates that the parameters taking adaptive updates are dominant, which is expected because the adaptive optimizers like AdamW are preferred in transformer. As indicated in Figure 4c, most parameters are updating stochastically, which explains why AdaDQH has a similar curve to SGD in Figure 3b before the 100th epoch. The ratio grows from 3% to 5% afterwards, resulting in a better PPL in the fine-tuning stage. As for Figure 4d, the model of the RecSys task is training for only one epoch, and AdaDQH gradually switch to the adaptive updates for a better fit to the data.

## 4 CONCLUSION

In this paper, we propose the AdaDQH optimizer, which can evolve from stochastic to adaptive by auto switch of the precondition matrix and has better generalization compared to the SOTA optimizers. We theoretically prove the convergence rate in both convex and non-convex stochastic settings and conduct empirical evaluation in real-world datasets of different scenes. The results clearly demonstrate the advantages of our optimizer in getting significantly better performance. Finally, we analyze how it is able to auto switch from stochastic to adaptive, and the rigorous effect of the hyperparameter $\delta$ which controls the auto-switch process.

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
