# OpenReview forum: "AdaDQH Optimizer: Evolving from Stochastic to Adaptive by Auto Switch of Precondition Matrix"
_ICLR.cc/2023/Conference — Submitted to ICLR 2023_

### Official Review · Reviewer_cyVQ · 2022-10-22

**Confidence:** 5
**Clarity, Quality, Novelty And Reproducibility:** 1. The research is a novelty since it…
**Correctness:** 4
**Technical Novelty And Significance:** 3
**Empirical Novelty And Significance:** 3
**Recommendation:** 8

**Strength And Weaknesses:**

1. The paper provides mathematical details to support time-complexity and computational analysis, and bounds for establishing convergence.
2. Results for the paper cover problems from different areas in Machine Learning and Optimization, adding to credibility and usefulness in the future.
3. An essential point covered in the paper is: Trade-off of computations, time and generalization.
4. It would be interesting to note the performance of AdaDQH on highly non-convex optimization problems  (i.e. functions with several local minima) like Ackley and Schwefel. This would help test the idea of whether AdaDQH is able to converge to global minima or not. Challenge problems from CEC 2019 would also help.
5. An important question arises: Is AdaDQH able to converge to global minima of few functions wherein the current state-of-the-art algorithms are only able to make it to local minima? This would make it a significant improvement over the current state-of-the-art algorithms.

**Summary Of The Paper:**

The research is on developing an adaptive optimization algorithm called AdaDQH, which proposes a new technique to approximate the Hessian matrix in the second-order gradient descent scheme in a computationally efficient manner. The authors demonstrate the optimizer’s performance on datasets from Natural Language Processing, Computer Vision, and Recommendation Systems to show faster convergence and better generalization. The paper also provides theoretical proofs on bounds on regret and time complexity to explain the trade-off between generalization, time and computational resources. Weights in a deep-network at iteration are updated as:
   w_(t+1) = w_t − α_t B_t^(−1t) m_t
where,α_t is the step-size, represents a linear combination of the gradients ∇_wL(w_t) , and B_t, called as the precondition matrix is an approximation of the Hessian H_t . Several previous second-order optimization algorithms like AdamW, AdaBelief, AdaHessian have developed methods to approximate . The current algorithm, AdaDQH, selects B_t as: B_t^2 = diag(EMA(s_1s_1^T, s_2s_2^T, s_ts_t^T ))/(1 − β_2^t))
where, EMA represents an exponential moving average, and is given by a recurrent relation in terms of s_t.
The structure of is adopted for bias-correction is similar to Adam. In addition to approximating the Hessian, the research also proposes a way to switch between an adaptive version and EMA version for the gradients.



**Summary Of The Review:**

Research paper is well-presented and produces a novel technique to approximate the Hessian for optimization. Overall, it’s an important paper to be considered in Machine Learning.

Additional Comments:

What is s_t, b_t and beta_2?
Could different notations be used for vector and scalar?
The paper updates B_t with max(B_t , delta*I). Why is this so?
How was equation 3 obtained from equation 2?
Is it possible to give mathematical guarantees on autoswitch? Say when it switches and in which conditions does it occur.

---

> ### Author Response · Authors · 2022-11-12
> **Thanks for your encouraging review. We have addressed all your concerns.**
>
> Thanks for your encouraging and valuable comments. We will explain your concerns point by point.
>
> **Q1: It would be interesting to note the performance of AdaDQH on highly non-convex optimization problems (i.e. functions with several local minima) like Ackley and Schwefel. This would help test the idea of whether AdaDQH is able to converge to global minima or not. Challenge problems from CEC 2019 would also help. Is AdaDQH able to converge to global minima of few functions wherein the current state-of-the-art algorithms are only able to make it to local minima? This would make it a significant improvement over the current state-of-the-art algorithms.**
>
> It's a very interesting and challenging question. When it comes to highly non-convex optimization problems, no one can guarantee a global minima convergence. However, AdaDQH can take advantage of both the stochastic and the adaptive methods, due to its auto-switch mechanism. For example, we test SGDM, Adam and AdaDQH on Ackley function, and the trajectories are shown in `rebuttal/Ackley.pdf` of the supplementary material. Both SGDM and AdaDQH can converge to the global minima while Adam gets stuck in local minima. All three optimizers are using the same initial learning rate and scheduler, though it may vary for different initial points. Another interesting observation is that AdaDQH (green line) tend to travel along one axis and switch to another, taking a shorter path than SGDM. We believe it also benefits from its unique auto-switch design, but due to time limitation, we'll leave it for future work.
>
> **Q2.1: What is s_t, b_t and beta_2?**
>
> $s_t$ is the Hessian approximation in step $t$ which is defined in line 5 of Algorithm 1.
>
> $b_t$ is EMA of $s_t^2$ for reducing the estimated variance which is defined in line 6 of Algorithm 1, i.e., the diagonal elements of $B_t^2$ where $B_t$ is the precondition matrix.
>
> $\beta_2$ is the exponential decay rate of EMA of $s_t^2$.
>
> **Q2.2: Could different notations be used for vector and scalar?**
>
> We use boldface to distinguish vectors from scalars in the revision.
>
> **Q2.3: The paper updates B_t with max(B_t , delta*I). Why is this so?**
>
> The choice of optimizer for specific task is becoming an essential hyperparameter, e.g. SGD for CNN and AdamW for Transformers. Intuitively, we want to design a general one which does well across tasks. Above that, instead of simply combining the stochastic and the adaptive, the optimizer ought to choose from both sides during training process per parameter, where the $\max (B_t, \delta*\mathbb{I})$ comes in. When the diagonal elements of $B_t$ are relatively larger than $\delta$, AdaDQH takes a confident step in the adaptive way. Otherwise, the update is EMA, i.e. $m_t$, with a constant scale ${\alpha_t}/({1-\beta_1^t})$, similar to SGD with momentum.
>
> **Q2.4: How was equation 3 obtained from equation 2?**
>
> Thank you for pointing out this issue. In Eq. (3), we use $m_t$ to replace $\nabla{}f(w_t)$ since it is ineffective to calculate $\nabla{}f(w_t)$ when the training set is large. $m_t$ is a kind of approximation of the expectation of $g_t$, i.e., $m_t\approx\mathbf{E}[g_t] = \nabla{}f(w_t)$. We have added the explanation into the revision to fix the issue.
>
> **Q2.5: Is it possible to give mathematical guarantees on autoswitch? Say when it switches and in which conditions does it occur.**
>
> As shown in Section 3.5, we know that the effect of the auto-switch hyperparameter $\delta$ depends on different scenes and tasks, i.e., depending on the distribution of the data. As far as we know, there are no powerful mathematical tools that are capable of depicting data distribution in high dimensions. Therefore, it is challenging to provide a mathematical analysis on when it would be beneficial to switch.
>
> ----
> Again we thank the reviewer for the valuable feedback.

---

> > ### Comment · Reviewer_cyVQ · 2022-11-20
> > **Author response**
> >
> > Thank you for answering my questions with confidence and clarity. My ratings remain the same.

---

### Official Review · Reviewer_EHjo · 2022-10-24

**Confidence:** 4
**Correctness:** 3
**Technical Novelty And Significance:** 2
**Empirical Novelty And Significance:** 3
**Recommendation:** 5

**Clarity, Quality, Novelty And Reproducibility:**

The idea is easy to follow and well presented.

Novelty can be found in W1.

Code is provided for reproducibility.

**Strength And Weaknesses:**

**Strength:**
- S1. The idea to approximate Hessian is interesting.
- S2. On tested tasks, AdaDQH exhibits improved performance.

**Weakness:**

- W1. (Novelty.) Two of key ideas, Hessian approximation, and auto switch have been used in the literatures but not well cited. For example, the idea of using gradient difference to estimate Hessian appeared in http://proceedings.mlr.press/v70/zheng17b.html, and the auto switch shares similar idea to https://arxiv.org/abs/1902.09843. In addition, although the idea of approximating Hessian is interesting, the motivating equation in two lines after Alg. 1 is less convincing. First, the value of $\theta$ is taken as 1, however, the reason is unclear. If it is taken from mean value theorem, the choice of $\theta$ can be different in different cases. Second, the reason to set $\Delta w$ to $e_i$ is also unclear, and it looks like this is not the case when updating the model weights. It can be more reasonable if the authors can plot the difference between the proposed Hessian estimation and real Hessian. Another question is that how well does EMA approximates the true gradient. The lack of justification makes the claim "better estimated Hessian" less convincing.

- W2. (Numerical experiments.) It looks like the models for evaluating the proposed algorithm are not well chosen. For example, it is standard to use ResNet50 rather than ResNet18 on ImageNet. And ResNet18 can easily achieve 93 - 94\% test accuracy on CIFAR10, while the tested ResNet20 and ResNet32 struggle on the same task. The choice of testing models makes the numerical results less competitive.
And why are some results in Table 4 are training on P100 and some on V100?


(Minor 1) Does the experiment in Figure 1 take stochastic gradient into account?

(Minor 2) The theory uses diminishing $\beta_{1,t}$ while the main algorithm adopts fixed $\beta_1$.


**Summary Of The Paper:**

This paper studies an Adam type optimizer, adaDQH, by designing a new rule to update $v$. The idea is i) using gradient difference to approximate (diagnalized) Hessian, and ii) using a max in the denominator of to switch AdaDQH to SGD when necessary. Theoretical results and numerical tests are carried out to support the convergence of proposed AdaDQH.

**Summary Of The Review:**

The proposed AdaDQH is interesting in how the Hessian is estimated. However, this paper does not provide convincing explanation to their Hessian estimate, and does not compare with existing works with similar ideas. Some of experiments are not carried out on standard models making their numerical improvement less impressive.

---

> ### Author Response · Authors · 2022-11-12
> **Thanks for your thoughtful review. We have addressed all your concerns. (Part 1 / 3)**
>
> Thanks for your thoughtful and valuable comments. We will explain your concerns point by point.
>
> **Q1.1: Two of key ideas, Hessian approximation, and auto switch have been used in the literatures but not well cited. For example, the idea of using gradient difference to estimate Hessian appeared in http://proceedings.mlr.press/v70/zheng17b.html, and the auto switch shares similar idea to https://arxiv.org/abs/1902.09843.**
>
> The first paper [1] you mentioned employs the diagonal elements of empirical Fisher information to approximate Hessian, rather than gradient difference.
>
> The second paper [2] adopts clipping on the learning rate of Adam, whose upper and lower bounds are non-increasing and non-decreasing functions, respectively, which would converge to the learning rate of SGD. The same technique is also mentioned in [3], whose upper and lower bounds are constants. The purpose of this method is to switch an adaptive optimizer to SGD for better generalization performance. We have supplemented these works into the related work section of the revision. However, the design of the auto-switch for AdaDQH is completely different from AdaBound [2]. We do not expect AdaDQH to perform adaptive optimization at the early stages and switch to SGD at later stages, but it should be able to switch freely between stochastic and adaptive. As shown in Section 3.5, AdaDQH behaves more like SGD at the early stage and switches to adaptive for fine-tuning in the CNN CV task, while it takes adaptive updates dominantly all the time in the Transformer NLP task. Therefore, the design of AdaDQH can be more flexible than AdaBound. According to [4] and [5], AdaDQH outperforms AdaBound across different experiments, as listed below.
> | Optimizer | 3-Layer LSTM, test PPL (lower is better) | ResNet18 on ImageNet, Top1 accuracy |
> |----|----|----|
> | AdaBound | 63.60 [4] | 68.13 (100 epochs) [5] |
> | AdaDQH | 60.89 (better) | 70.19 (90 epochs, still better) |
>
> **Q1.2: In addition, although the idea of approximating Hessian is interesting, the motivating equation in two lines after Alg. 1 is less convincing. First, the value of θ is taken as 1, however, the reason is unclear. If it is taken from mean value theorem, the choice of θ can be different in different cases. Second, the reason to set Δw to ei is also unclear, and it looks like this is not the case when updating the model weights. It can be more reasonable if the authors can plot the difference between the proposed Hessian estimation and real Hessian.**
>
> We can rewrite the first approximation of Eq. (5) of the revision as follows:
> $$\mathbb{E}[g_{t, i} - g_{t-1,i}] = \nabla\nabla_{i}f(w_t)\cdot\Delta{}w + \underbrace{\left(\nabla\nabla_{i}f(w_{t-1}+\theta\Delta{}w) - \nabla\nabla_{i}f(w_t)\right)\cdot\Delta{}w}_{h}.$$
>
> Assume $\nabla_{i}f$ is $L$-smooth, $\forall{}i$, i.e., $\\|\nabla\nabla_{i}f(w_t) - \nabla\nabla_{i}f(w_{t-1})\\|\leq L\\|w_t - w_{t-1}\\|,\ \forall{}i$, then we have
> $$\begin{align}
> \\|h\\|\leq\\|\nabla\nabla_{i}f(w_{t-1}+\theta\Delta{}w) - \nabla\nabla_{i}f(w_t)\\|\\|\Delta{}w\\|
> \leq{}L\\|w_{t-1} + \theta\Delta{}w - w_t\\|\\|\Delta{}w\\|=L|\theta - 1|\\|\Delta{}w\\|^2.
> \end{align}$$
>
> Hence, $h$ can be kept small when $\\|\Delta{}w\\|$ is small. Otherwise, we can add a constraint to make sure $\\|\Delta{}w\\|$ is small, though we don't do this in practice. That's the reason we can take $\theta=1$ to approximate $\mathbb{E}[g_{t, i} - g_{t-1,i}]$.
>
> The second approximation of Eq. (5) comes from two steps. First, we employ a widely used diagonalization trick to approximate $\nabla\nabla_{i}f(w_t)\cdot\Delta{}w$, i.e., $\left[ \nabla\nabla_{1}f(w_t)\cdot\Delta{}w, \dots, \nabla\nabla_{n}f(w_t)\cdot\Delta{}w\right]^T=\nabla^2f(w_t)\Delta{}w \approx \text{diag}(\nabla^2f(w_t))\Delta{}w$, hence, $\nabla\nabla_{i}f(w_t)\cdot\Delta{}w\approx\nabla_{i}\nabla_{i}f(w_t)\Delta{}w_i$. We can use $\mathbb{E}[g_{t, i} - g_{t-1,i}] / \Delta{}w_{i}$ to estimate the $i$-th diagonal element of Hessian. However, the estimation could be very unstable if $\Delta{}w_i$ is too small. Thus, in the second step, we normalize $w_t$ such that $|\Delta{}w_i| = 1$. (Here we ignore $\Delta{}w_i = -1$ since the update direction is wholly decided by $m_t$) It seems to contradict the condition of the first approximation which needs $\\|\Delta{}w\\|$ is small. In fact, we can normalize $w_t$ such that $|\Delta{}w_i| = \epsilon$ where $\epsilon$ is small and $\epsilon$ can be incorporated into the learning rate. Therefore, we use the notation $\Delta{}w\approx{}e_i$ for simplicity.
>
> It is challenging to depict the difference between the proposed Hessian estimation and real Hessian since the function can be arbitrary. For example, if $f(x, y) = x^2 + y^2$, let the exponential decay rates $\beta_1 = \beta_2 = 0$ and learning rate = $1$, then the Hessian estimation equals to the real Hessian.

---

> > ### Author Response · Authors · 2022-11-12
> > **Thanks for your thoughtful review. We have addressed all your concerns. (Part 2 / 3)**
> >
> > (follow to https://openreview.net/forum?id=W-VfwHzA2yg&noteId=N3tHMQ6aTSu)
> >
> > **Q1.3: Another question is that how well does EMA approximates the true gradient.**
> >
> > Using the EMA of stochastic gradient to approximate the true gradient has become a widely used method in adaptive optimizers.
> > From the recursion formula, $m_t = \beta_1 m_{t-1} + (1-\beta_1)g_t$ with $m_0 = 0$, we can obtain
> > $m_t = (1-\beta_1)\sum_{i=1}^{t} \beta_1^{t-i} g_i$. Taking expectations of the left-hand and right-hand sides, we have
> > $$\mathbb{E}(m_t) = \mathbb{E}\left[(1-\beta_1)\sum_{i=1}^{t} \beta_1^{t-i} g_i \right] = \mathbb{E}[g_t] \cdot (1-\beta_1^t) + \zeta,$$
> > where $\zeta = 0$ if the true first moment $\mathbb{E}[g_t]$, i.e., true gradient, is stationary; otherwise $\zeta$ can be kept small since EMA assigns small weights to gradients too far in the past by choosing $\beta_1$. The similar analysis can be found in [8].
> >
> > Now assume $\mathbb{E}[g_t]$ and $\mathbf{Var}[g_t]$ are stationary and $\beta_1\in(0, 1)$, then we have
> > $$\mathbf{Var}\left[\frac{m_t}{1-\beta_1^t}\right] = \frac{1}{(1 - \beta_1^t)^2}\mathbf{Var}\left[(1-\beta_1)\sum_{i=1}^{t} \beta_1^{t-i} g_i \right] = \frac{(1+\beta_1^t)(1-\beta_1)}{(1-\beta_1^t)(1+\beta_1)}\mathbf{Var}[g_t] < \mathbf{Var}[g_t].$$
> > Therefore, $m_t/(1-\beta_1^t)$ is the unbiased estimation of $\nabla{}f(w_t)$ and has lower variance than $g_t$.
> >
> > **Q1.4: The lack of justification makes the claim "better estimated Hessian" less convincing.**
> >
> > See the responses to Q1.2 and Q1.3.
> >
> > **Q2.1: It looks like the models for evaluating the proposed algorithm are not well chosen. For example, it is standard to use ResNet50 rather than ResNet18 on ImageNet. And ResNet18 can easily achieve 93 - 94% test accuracy on CIFAR10, while the tested ResNet20 and ResNet32 struggle on the same task. The choice of testing models makes the numerical results less competitive.**
> >
> > In both AdaBelief [4] and AdaHessian [6] paper, ResNet18 is chosen as a standard model to be evaluated on ImageNet. In order to make a better comparison with them, we also used ResNet18 on ImageNet. Besides, following [6], we choose ResNet20 and ResNet32 for the experiments on Cifar10.
> >
> > We follow the naming convention from the original ResNet paper [7]. The ResNets designed for ImageNet and Cifar10 differ in in-layer map size and filters, which leads to confusion. We emphasize that ResNet18 (designed for ImageNet) is much more complex than ResNet20 and ResNet32 (designed for Cifar10), as the model size (listed in Table 2) for ResNet18, ResNet20 and ResNet32 is 11.69M, 0.27M and 0.47M, respectively. More details can be found in [7]. Hence, it is quite understandable that the test accuracy of ResNet20 and ResNet32 is lower than that of ResNet18. When training a ResNet18 on Cifar10, we can easily get 95% test accuracy on Cifar10 in our experimental configuration. Moreover, as we mentioned in Appendix E.5, AdaDQH can achieve 95.79% accuracy for ResNet18 when using the same training configuration as the experiment of SGD in [9], which exceeds the current SOTA result (see https://paperswithcode.com/sota/stochastic-optimization-on-cifar-10-resnet-18).
> >
> > **Q2.2: Why are some results in Table 4 are training on P100 and some on V100?**
> >
> > We have access to a pool of mixed GPUs in our internal production system, and we utilize them as much as we can.
> >
> > **Q3: Does the experiment in Figure 1 take stochastic gradient into account?**
> >
> > No. The test functions are optimized deterministically.
> >
> > **Q4: The theory uses diminishing $\beta_{1,t}$ while the main algorithm adopts fixed $\beta_1$.**
> >
> > We improve the bound of AdaDQH in non-convex stochastic settings (Theorem 2) and prove the convergence when $\beta_{1,t}$ is a constant (Corollary 3), i.e., $\beta_{1,t}$ is only needed to be a non-increasing function. These modifications have been incorporated into the revision.
> >
> > ----
> > We thank you again for your valuable feedback. If our response has been satisfactory, we would really appreciate it if you would consider updating your score.

---

> > > ### Author Response · Authors · 2022-11-12
> > > **Thanks for your thoughtful review. We have addressed all your concerns. (Part 3 / 3)**
> > >
> > > References:
> > >
> > > [1] Shuxin Zheng et al., Asynchronous Stochastic Gradient Descent with Delay Compensation. ICML 2017.
> > >
> > > [2] Liangchen Luo et al., Adaptive gradient methods with dynamic bound of learning rate. ICLR 2019.
> > >
> > > [3] Nitish Shirish Keskar and Richard Socher. Improving generalization performance by switching from adam to SGD. CoRR, abs/1712.07628, 2017.
> > >
> > > [4] Juntang Zhuang et al., Adabelief optimizer: Adapting stepsizes by the belief in observed gradients. NIPS 2020. https://github.com/juntang-zhuang/Adabelief-Optimizer/.
> > >
> > > [5] Jinghui Chen and Quanquan Gu, "Closing the generalization gap of adaptive gradient methods in training deep neural networks".
> > >
> > > [6] Yao, Z. et al., ADAHESSIAN: An Adaptive Second Order Optimizer for Machine Learning. AAAI 2021.
> > >
> > > [7] He, K., Zhang, X., Ren, S. & Sun, J. Deep Residual Learning for Image Recognition. CVPR 2016.
> > >
> > > [8] Diederik P. Kingma and Jimmy Lei Ba. Adam: A method for stochastic optimization. ICLR 2015.
> > >
> > > [9] Moreau, T. et al., Benchopt: Reproducible, efficient and collaborative optimization benchmarks. 2022.

---

> > > > ### Comment · Reviewer_EHjo · 2022-11-26
> > > > **Thanks for the detailed responses**
> > > >
> > > > Thank you coping with my questions. Some of them are addressed nicely, and here are some follow-ups.
> > > >
> > > > Q1.1 Please see eqn (5) in [1], and rearrage it to see how gradient difference is used to estimate Hessian.
> > > >
> > > > Q1.2 While the first approximation makes sense, the second approximation is still not clear yet. It looks like the second approximation works the best when $|| w_t - w_{t-1}||$ is small. Let's take a strongly convex problem as an example to get some intuition. Suppose that $w_0$ is intialized far away from the optimal solution. In order to converge fast, one might want to have  $|| w_t - w_{t-1}||$ not too small so that it is possible to get to $w^*$ with less iterations. However, this is in contrast with the need of an accurate approximation.
> > > >
> > > > In addition, if the authors can cope with an e.g., logistic regression problem, and plot the Hessian estimate quality under the default parameter selection, the 'Hessian approximation' can be more convincing.
> > > >
> > > > Q1.3 I am not sure on understanding the derivations here. For example, in the first equation, why is $\sum_i \beta_i^{t-i} g_i = (1 - \beta_1^t) g_t$? In particualr, can the authors help me with changing $g_i$ to $g_t$? The derivation on the variance is also not clear to me for the same reason.
> > > >
> > > >  Q2. It is more fair to compare #params when using the same dataset.. For example, Cifar10 and ImageNet have 10 and 1000 classes, respectively. This will make at least the number of parameters in the softmax layer differs a lot.

---

> > > > > ### Author Response · Authors · 2022-11-29
> > > > > **Thanks for your thoughtful review. We have addressed your additional concerns.**
> > > > >
> > > > > **Q1: Please see eqn (5) in [1], and rearrange it to see how gradient difference is used to estimate Hessian.**
> > > > >
> > > > > Thank you for pointing this out, and we apologize for ignoring this part. The similar idea is also appeared in much earlier work, such as LBFGS [1] (which is mentioned in our paper). It can be summarized as the so-called quasi-Newton rule, i.e., choosing $A_{t+1}$ such that $A_{t+1}(g_{t+1} - g_t) = w_{t+1} - w_t$, where $A_{t+1}$ can be seen as the inverse of the Hessian. We will consider adding more introduction of this original idea in the latter version.
> > > > >
> > > > > We emphasize that despite the difference of two gradients to estimate Hessian is well known, there has not been any work that demonstrated the efficacy of this Hessian approximation on optimization methods in the past several years. This suggests that making the seemingly simple idea work is non-trivial. In fact, except for the Hessian approximation, auto-switch plays a critical role. The two design parts work together to achieve better performance across different tasks.
> > > > >
> > > > > **Q2.1: While the first approximation makes sense, the second approximation is still not clear yet. It looks like the second approximation works the best when $\\|w_t−w_{t−1}\\|$ is small. Let's take a strongly convex problem as an example to get some intuition. Suppose that $w_0$ is initialized far away from the optimal solution. In order to converge fast, one might want to have $\\|w_t−w_{t−1}\\|$ not too small so that it is possible to get to $w^{*}$ with less iterations. However, this is in contrast with the need of an accurate approximation.**
> > > > >
> > > > > As explained in the response of Q1.2 of https://openreview.net/forum?id=W-VfwHzA2yg&noteId=N3tHMQ6aTSu, $\\|w_t - w_{t-1}\\|$ has been incorporated into the learning rate. The learning rate can be relative large in the early stage training, which means $w_t$ is far from the optimal point $w^*$; while become small when $w_t$ is close to $w^*$.
> > > > >
> > > > > **Q2.2: In addition, if the authors can cope with an e.g., logistic regression problem, and plot the Hessian estimate quality under the default parameter selection, the 'Hessian approximation' can be more convincing.**
> > > > >
> > > > > We generate a sample dataset using normal distribution, centered in (0, 0) and (-2, -2) for label 0 and 1, respectively. We plot $\mathrm{diag}(|H|)$ (the diagonal of the hessian of $\theta$), and its corresponding $|s_t|$ (AdaDQH's approximation) over epochs. As shown in https://imgur.com/a/xPcwkaF, the approximation's direction correctly follows the hessian in logistic regression settings.
> > > > >
> > > > > **Q3: I am not sure on understanding the derivations here. For example, in the first equation, why is $\sum_{i}\beta_{i}^{t-i}g_i = (1 - \beta_1^t)g_t$? In particular, can the authors help me with changing $g_i$ to $g_t$? The derivation on the variance is also not clear to me for the same reason.**
> > > > >
> > > > > From the response of Q1.3 of https://openreview.net/forum?id=W-VfwHzA2yg&noteId=vF_3FJK1N2j, we have
> > > > > $$\mathbb{E}(m_t) = \mathbb{E}\left[(1-\beta_1)\sum_{i=1}^{t} \beta_1^{t-i} g_i \right] = \mathbb{E}[g_t] \cdot (1-\beta_1^t) + \underbrace{(1-\beta_1)\sum_{i=1}^{t} \beta_1^{t-i}\left(\mathbb{E}[g_i] - \mathbb{E}[g_t]\right)}_{\zeta}.$$
> > > > > If $\mathbb{E}[g_t]$ is stationary, i.e., $\mathbb{E}[g_i] = \mathbb{E}[g_t],\ \forall{}i\leq{}t$, we have $\zeta = 0$; otherwise $\zeta$ can be kept small since EMA assigns small weights to gradients too far in the past by choosing $\beta_1$. The derivation on the variance is the same since we assume $\mathbf{Var}[g_t]$ is stationary.
> > > > >
> > > > > **Q4: It is more fair to compare #params when using the same dataset. For example, Cifar10 and ImageNet have 10 and 1000 classes, respectively. This will make at least the number of parameters in the softmax layer differs a lot.**
> > > > >
> > > > > We list the parameters of models used in order to avoid the ResNet naming confusions. As for ResNet, the size and number of filters, the size of feature maps of a certain block are identical only for the same dataset. Originally, ResNet18 arch is proposed for ImageNet, while following works apply the arch on Cifar10 and can easily achieve 93 - 94% test accuracy. However, ResNet20 and ResNet32 are originally proposed for Cifar10, which is much lighter than the ResNet18 mentioned, though they have more layers.
> > > > >
> > > > > Furthermore, we design the experiments based on the previous related papers for reproduction and comparison.
> > > > >
> > > > > -----
> > > > > We hope that this answer your questions, please do not hesitate if you require any additional information.
> > > > >
> > > > > Reference:
> > > > >
> > > > > [1] Richard H. Byrd et al., A limited memory algorithm for bound constrained optimization. SIAM J. Sci. Comput 1995.

---

### Official Review · Reviewer_FWdz · 2022-10-24

**Confidence:** 3
**Clarity, Quality, Novelty And Reproducibility:** 1. The proposed method ADADQH is in f…
**Correctness:** 3
**Technical Novelty And Significance:** 2
**Empirical Novelty And Significance:** 1
**Recommendation:** 5

**Strength And Weaknesses:**

Strength:
1. The paper propose new types of modifications to AdaBelief, aiming to improve its performance, which I haven't seen from literature before.
2. Theoretical analysis is provided for the new method.

Weekness:
(1) As I mentioned earlier, ADADQH made two modifications to AdaBelief. It is not clear which modification plays a crucial role for the performance gain. I am guessing the first modification for computing s_t makes a difference in performance improvement. The reason is that m_t/(1-beta_1^t)-m_{t-1}/(1-beta_1^{t-1}) = (1-beta_1)/(1-beta_1^t) * [g_t - m_{t-1}/(1-beta_1^{t-1})]. The scalar  (1-beta_1)/(1-beta_1^t) essentially reduce the variance of the adaptive stepsizes in comparison to AdaBelief which computes the EMA of (g_t-m_t)^2.  In a recent paper "On Exploiting Layerwise Gradient Statistics for Effective Training of Deep Neural Networks", another method named Aida was proposed to reduce the variance of the adaptive stepsizes of AdaBelief by performing vector projections between m_t and g_t before being used for the computing the 2nd momentum. I suggest the authors to conduct performance also in comparison to Aida. In the abstract, the authors state that ADADQH is compared to SOTA optimizers. Both Aida and ADADQH are closely related to AdaBelief and should be compared.

(2) In AdaBelief paper, the validation accuracy of AdaBelief for training ResNet34 over CIFAR10 is above 95%. In the current paper, the validation accuracy of AdaBelief for training ResNet32 over CIFAR10 is below 93%. Even though ResNet 34 is slightly complicated than ResNet32, I think the performance gap is too large for AdaBelief.  I suggest the authors take the opensource from AdaBelief to make a direct comparison.

(3) My personal experience is the selection of parameter epsilon in Adam and AdaBelief plays a very important role in the performance. In order to make a fair comparison, the parameter epsilon needs to searched for each optimizer. In Figure 1, I don't see that the parameter epsilon  is being searched for Adam and AdaBelief. Inproper epsilon value in Adam and AdaBelief would slow down their convergence speed.

**Summary Of The Paper:**

The paper extends AdaBelief by making two modifications: (1) the 2nd momentum s_t is computed by taking the EMA of (m_t/(1-beta_1^t)-m_{t-1}/(1-beta_1^{t-1}) )^2 in comparison to the EMA of (m_t-g_t)^2 in AdaBelief; (2) the parameter delta (or epsilon  in AdaBelief) is introduced in a slightly different manner to avoid division by zero.  The new optimiser is referred to as ADADQH. Experiments on different DNN tasks indicates that ADADQH produce either better and competitive performance.

**Summary Of The Review:**

The paper proposes two new modifications to AdaBelief, and demonstrates performance gain on both synthetic and real-data experiments. 1. The connection between the proposed method  ADADQH and the two existing methods AdaBelief and Aida should be studied, or properly described in the introduction in the paper so that the readers better understand the reasoning behind it. 2. The parameter epsilon needs to be searched for each optimizer in order to make a fair comparison in all experiments. 3. The paper needs to conduct ablation study to find out which modification in ADADQH is more important to improve performance in AdaBelief.

---

> ### Author Response · Authors · 2022-11-12
> **Thanks for your constructive review. We have addressed all your concerns. (Part 1 / 2)**
>
> Thanks for your constructive and valuable comments. We will explain your concerns point by point.
>
> **Q1.1: As I mentioned earlier, ADADQH made two modifications to AdaBelief. It is not clear which modification plays a crucial role for the performance gain.**
>
> As shown in Section 3.5, we note that the effect of the auto-switch hyperparameter $\delta$ depends on different scenes and tasks. Based on our results in Fig 4(b),4(c), each modification plays the major role respectively, which both leads to a performance gain.
>
> **Q1.2: Both Aida and ADADQH are closely related to AdaBelief and should be compared.**
>
> In our paper, we have compared the performance of AdaDQH and AdaBelief. As for Aida, we note that the Aida paper [4] is not officially published. Therefore, we do not think it is necessary to mention Aida in our article. But for your concern, we will compare the performance of AdaDQH and Aida here. The results are shown below. We carried out the experiments for training ResNet34 over Cifar10 with the same training configurations as in Aida article [4] and exactly replicated the results of Aida. AdaDQH has better performance than Aida as expected.
>
> | Optimizer |  Accuracy |
> |----|----|
> | Aida’s result we calculated |	0.9537 $\pm$ 0.0008 |
> | Aida’s result from Aida's article [4] | 0.9542 $\pm$ 0.0013 |
> | AdaDQH | 0.9566 $\pm$ 0.0007 |
>
> **Q2: In AdaBelief paper, the validation accuracy of AdaBelief for training ResNet34 over CIFAR10 is above 95%. In the current paper, the validation accuracy of AdaBelief for training ResNet32 over CIFAR10 is below 93%. Even though ResNet 34 is slightly complicated than ResNet32, I think the performance gap is too large for AdaBelief. I suggest the authors take the opensource from AdaBelief to make a direct comparison.**
>
> We follow the naming convention from the original ResNet paper [1]. The AdaBelief paper uses ResNet34 (designed for ImageNet, 21.28M in model size) to train over Cifar10, which is much more complex than ResNet32 (designed for Cifar10, 0.47M in model size) we use.
>
> For your information, we compare the performance of AdaDQH and AdaBelief with the same training configuration as in AdaBelief's paper. The results are shown below. AdaDQH achieves comparable performance with AdaBelief in that case.
>
> | Optimizer |  Accuracy |
> |----|----|
> | AdaBelief | 0.9539 $\pm$ 0.0009|
> | AdaDQH | 0.9539 $\pm$ 0.0010|
>
> **Q3: My personal experience is the selection of parameter epsilon in Adam and AdaBelief plays a very important role in the performance. In order to make a fair comparison, the parameter epsilon needs to searched for each optimizer. In Figure 1, I don't see that the parameter epsilon is being searched for Adam and AdaBelief. Inproper epsilon value in Adam and AdaBelief would slow down their convergence speed.**
>
> Figure 1 mainly shows the convergence speed on toy examples, where we use the corresponding default epsilon or delta value for all optimizers including AdaDQH. We also conduct experiments with different epsilon or delta values, but find no significant influence on convergence speed. However, learning rate is the major factor in toy examples, as shown in Figure 5 in Appendix E.1.
>
> For other experiments on real-world datasets, epsilon for AdaBelief [2] is carefully tuned on various datasets and models, and we use the best parameters recommended. For Adam, we conduct experiments on epsilon selection, shown in Figure 6(b) in Appendix E.6. Epsilon equal to 1e-8 turns out to be a proper choice for Adam, and it enables us to reproduce the Adam results provided in other papers, like AdaHessian [3] and AdaBelief [2].
>
> **Q4: In the paper, I don't see the study or discussion on the connection between AdaBelief and ADADQH. The authors only mention AdaBelief without detailed information.**
>
> The precondition matrix $B_t$ of AdaBelief is $\sqrt{\text{Var}(g_t)}$, i.e., $g_t - m_t = \frac{1}{1 - \beta_{1}}m_t - \frac{\beta_1}{1 - \beta_1}m_{t-1} - m_t = \frac{\beta_1}{1-\beta_1}(m_t - m_{t-1})$. We can find that AdaBelief is the difference of the expectations without bias correction, which leads to its less approximation of the Hessian than AdaDQH.
>
> Our intuition of designing AdaDQH is how to approximate Hessian, not originated from AdaBelief. Thus, we omit the discussion on the connection between AdaDQH and AdaBelief in the paper.
>
>
> References:
>
> [1] He, K., Zhang, X., Ren, S. & Sun, J. Deep Residual Learning for Image Recognition. CVPR 2016.
>
> [2] Juntang Zhuang et al., Adabelief optimizer: Adapting stepsizes by the belief in observed gradients. NIPS 2020.
>
> [3] Yao, Z. et al., ADAHESSIAN: An Adaptive Second Order Optimizer for Machine Learning. AAAI 2021.
>
> [4] Zhang, G., Niwa, K. & Kleijn, W. B. On Exploiting Layerwise Gradient Statistics for Effective Training of Deep Neural Networks. 2022.

---

> > ### Author Response · Authors · 2022-11-12
> > **Thanks for your constructive review. We have addressed all your concerns. (Part 2 / 2)**
> >
> > (follow to https://openreview.net/forum?id=W-VfwHzA2yg&noteId=_jhe2Sz7Ox)
> >
> > **Q5: I don't get how (3) is obtained by solving (2). Note that m_t does not appear in (2). There is a gap between (2) and (3).**
> >
> > Thank you for pointing out this issue. In Eq. (3), we use $m_t$ to replace $\nabla{}f(w_t)$ since it is ineffective to calculate $\nabla{}f(w_t)$ when the training set is large. $m_t$ is a kind of approximation of the expectation of $g_t$, i.e., $m_t\approx\mathbb{E}[g_t] = \nabla{}f(w_t)$. We have added the explanation into the revision to fix the issue.
> >
> > -----
> > We thank you again for your valuable feedback. If you find our response adequate, we would greatly appreciate if you consider increasing your score.

---

### Author Response · Authors · 2022-11-12
**Rebuttal Revision**

We would like to thank all the reviewers for their constructive suggestions and useful insights. We have made the following changes in the freshly uploaded version:

- We improve the bound of AdaDQH in non-convex stochastic settings (Theorem 2) and prove the convergence when $\beta_{1,t}$ is a constant (Corollary 3).

- We add the explanation of using $m_t$ to replace $\nabla{}f(w_t)$ to clarify how Eq. (3) was obtained from Eq. (2).

- We add the introduction of the clipping technique of learning rate for switching an adaptive optimizer to SGD into the related work section.

- We use boldface to distinguish vectors from scalars.

---

### Decision · Program_Chairs · 2023-01-20

**Decision:**

Reject

**Justification For Why Not Higher Score:**

This is yet another optimization method as there are so many out there. Differences with existing ones are marginal and I don't see the publication of this work changing anyone's habit.

**Justification For Why Not Lower Score:**

N/A

**Metareview: Summary, Strengths And Weaknesses:**

This paper proposes a new optimizer based of Adam with a few modifications. The main question for this type of works is about the true goal: is it to provide a faster optimizer when all the hyperparameters are carefully tuned? Is it to provide an optimizer that is competitive across many different settings with the same value for the hyperparameters?

Because the reviewers barely engaged in a discussion with the authors, I had a closer look at the paper. As it is, I do not feel the theoretical part or the empirical part make a convincing enough argument. The Hessian approximation lacks validation, not in the sense that it does not work, but rather that many preconditioners end up giving good performance and I became wary of explanations.
As for the experimental part, most of the results are in the same ballpark as existing methods. I have yet to see cases where differences of that magnitude were enough to determine the choice of algorithm.

Ultimately, I could not be convinced of the impact and decided to side with both reviewers who gave a 5.